# Investigating the direct and indirect effects of forest fragmentation on plant functional diversity

Jenny Zambrano[1]*, Norbert J. Cordeiro[2,3], Carol Garzon-Lopez[4], Lauren Yeager[5], Claire Fortunel[6,7], Henry J. Ndangalasi[8], Noelle G. Beckman[9]

1 School of Biological Sciences, Washington State University, Pullman, Washington, United States of America, 2 Department of Biology (mc WB 816), Roosevelt University, Chicago, Illinois, United States of America, 3 Science & Education, The Field Museum, Chicago, Illinois, United States of America, 4 Grupo de Ecología y Fisiología Vegetal, Departamento de Ciencias biológicas, Universidad de los Andes, Bogotá, Colombia, 5 Department of Marine Science, University of Texas at Austin, Port Aransas, Texas, United States of America, 6 Department of Ecology and Evolutionary Biology, University of California, Los Angeles, California, United States of America, 7 AMAP (botAnique et Modélisation de l'Architecture des Plantes et des végétations), Université de Montpellier, CIRAD, CNRS, INRAE, IRD, Montpellier, France, 8 Botany Department, University of Dar es Salaam, Dar es Salaam, Tanzania, 9 Department of Biology & Ecology Center, Utah State University, Logan, Utah, United States of America

* jenny.zambrano@wsu.edu

**Data Availability Statement:** The data supporting the results are deposited in the The Knowledge Network for Biocomplexity repository (DOI: 10.5063/F1KS6PX9).

## Abstract

Ongoing habitat loss and fragmentation alter the functional diversity of forests. Generalising the magnitude of change in functional diversity of fragmented landscapes and its drivers is challenging because of the multiple scales at which landscape fragmentation takes place. Here we propose a multi-scale approach to determine whether fragmentation processes at the local and landscape scales are reducing functional diversity of trees in the East Usambara Mountains, Tanzania. We employ a structural equation modelling approach using five key plant traits (seed length, dispersal mode, shade tolerance, maximum tree height, and wood density) to better understand the functional responses of trees to fragmentation at multiple scales. Our results suggest both direct and indirect effects of forest fragmentation on tree functional richness, evenness and divergence. A reduction in fragment area appears to exacerbate the negative effects resulting from an increased amount of edge habitat and loss of shape complexity, further reducing richness and evenness of traits related to resource acquisition and favouring tree species with fast growth. As anthropogenic disturbances affect forests around the world, we advocate to include the direct and indirect effects of forest fragmentation processes to gain a better understanding of shifts in functional diversity that can inform future management efforts.

## Introduction

Forest loss and fragmentation result in long-lasting and complex changes in biodiversity that may go beyond the loss of species to include the alteration of functional diversity of remaining

**Funding:** JZ and NGB were supported by the National Socio-Environmental Synthesis Center under the US National Science Foundation (NSF) Grant DBI-1052875. CF benefited from an "Investissements d'Avenir" grant managed by Agence Nationale de la Recherche (CEBA, ref. ANR-10-LABX-25-01). Support for LY was also provided from NSF grant #OCE-1661683.

**Competing interests:** The authors have declared that no competing interests exist.

communities. Forest fragmentation threatens the long-term persistence of species [1–3], as well as the goods and services provided by those ecosystems [4]. Fragmentation is a hierarchical process that involves breaking apart the habitat of a focal species into populations isolated from each other in a matrix of modified habitat [5–7]. Changes in the spatial configuration of the landscape alter the abiotic and biotic filters that govern community assembly, selecting individuals with suites of traits that enable them to survive, grow, reproduce, and colonize remaining fragments. Species traits relate to physiological, morphological, and phenological functions [8–10], and local functional diversity can influence ecosystem functioning [11,12]. If species' traits in remaining fragments become more similar over time, a process known as functional homogenization, this could severely alter a variety of ecosystem functions performed by remaining communities and, by extension, the ecosystem services they provide. Previous studies provide evidence that forest fragmentation often favours plant species with traits within a specific range of values [e.g. 13], potentially leading to functional homogenization by reducing alpha diversity of functional traits [14]. By taking into account functional diversity within a community, we can better understand how species respond to fragmentation processes that alter the abiotic and biotic filters that govern community assembly.

Trait values that allow species to take advantage of recent disturbances are commonly hypothesised to determine species success in fragmented landscapes [15,16]. Recent studies have shown that reductions in fragment area and increases in the amount of edge habitat locally favour tree species with faster growth rates (e.g. pioneers), smaller seeds, shorter leaf life span and lower wood density [15–20]. Additionally, increased spatial isolation and an inhospitable matrix habitat are expected to select for abiotically-dispersed tree species and/or small-seeded, animal-dispersed tree species that have the potential for wide dissemination by attracting many seed-disperser species [16,18,20,21]. However, because of the variable results across studies and systems, there is limited consensus on the generality of the magnitude of these shifts and their drivers.

The process of landscape fragmentation can be considered at multiple, interacting scales. Fragmentation effects via fragment isolation or matrix quality that impact dispersal among fragments or meta-population dynamics may manifest most strongly at the landscape scale. In contrast, fragmentation effects via edge effects, fragment shape or size that impact fine-scale habitat quality and individual persistence may be best detected at the fragment-scale (as with our study with fragments ranging from 0.011–9.51 km$^2$). Furthermore, these landscape- and fragment-scale changes typically occur concurrently which may lead to interactions among various fragmentation effects. For example, dispersal between fragments typically declines with isolation and an inhospitable matrix habitat may exacerbate the effects of fragment isolation on species diversity [7]. In forest fragments, altered abiotic conditions such as greater desiccation through increased wind and light, causing higher temperatures and lower humidity, are among the main edge effects as the shape of fragments becomes narrower and/or as the size of fragments decreases [6]. Decreasing fragment size could both directly impact species persistence by lowering local population sizes and increasing edge effects as the relative amount of edge habitat is greater in smaller fragments. Teasing apart these co-occurring changes across spatial scales has posed a major challenge in predicting the net response of functional diversity to forest fragmentation to date [6,22,23].

Previous investigations have yielded mixed results, with functional diversity responding either positively or negatively to forest fragmentation [14,24,25]. This lack of consensus could be the result of not accounting for the direct and indirect effects of both landscape- and fragment-level effects. While measuring the independent effects of individual landscape properties is useful to identify mechanisms behind fragmentation-driven biodiversity changes, such approaches may miss critical indirect effects between fragment-level and landscape-level

fragmentation variables [26], and potentially leads to incorrect inferences and predictions regarding the impacts of forest fragmentation on communities. Structural Equation Models (SEM) have been proposed as an alternative tool to jointly study the direct and indirect effects of habitat amount and configuration because SEMs specify predictor variables that may not have been measured or that may be difficult to observe directly, and therefore measure the strength of causal relationships among predictors and provide rigorous estimates of direct and indirect effects [27].

Here, we use a SEM approach that permits the evaluation of direct and indirect effects of forest fragmentation on plant functional diversity in the East Usambara Mountains of Tanzania. This approach in particular allows us to tease apart the attributes of forest fragmentation that operate across different spatial scales and to compare the relative importance of local versus landscape-scale processes affecting different dimensions of functional diversity (e.g richness, evenness and divergence). In this study, we censused trees in plots across a fragmented rainforest in the East Usambara Mountains, an area in Africa known for its high levels of biodiversity and endemism [28] that is currently protected under the United Nations Educational, Scientific and Cultural Organization (UNESCO) Biosphere Reserve status. We hypothesize that:

1. The variation in functional diversity in response to fragmentation is mediated by both fragment- and landscape-scale factors (Fig 1). We expect that the impacts of fragment size, shape complexity, and edge effects on functional diversity are indirectly affected by landscape-level processes such as fragment isolation and matrix quality. For example, edge effects tend to be more severe in small and/or narrow or irregularly-shaped fragments, which would therefore affect functional diversity. Finally, we anticipate a greater negative effect of isolation on functional diversity for fragments surrounded by an inhospitable matrix habitat.

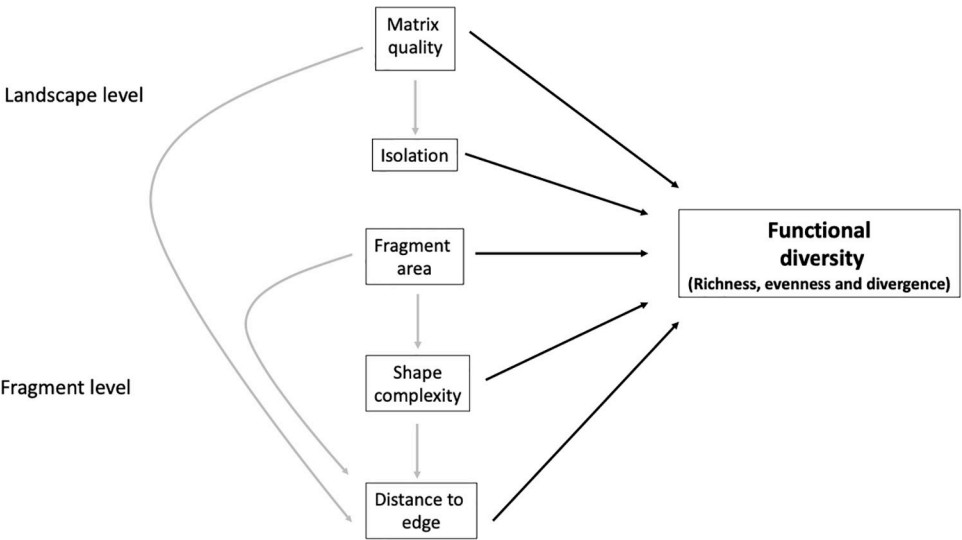

**Fig 1. Conceptual model illustrating the directional relationships between fragmentation processes occurring at the landscape and fragment level affecting functional diversity.** Functional diversity was defined in terms of functional richness, evenness and divergence. Functional metrics were fitted in separate models. Arrows indicate the hypothesized causal relationships, with dashed arrows representing indirect effects and continuous lines representing direct effects.

2. The effects of fragmentation are expected to impact functional diversity in several ways: a) functional richness, evenness and divergence of resource use traits are expected to decline with reduced fragment area and shape complexity as the amount of forest edge increases; b) low quality of matrix surrounding the remaining fragments is expected to exacerbate the environmental stress in edge habitats, further reducing functional richness, evenness and divergence (Fig 1); c) trait distribution is expected to become more skewed towards species with trait values associated with fast resource use (e.g. short stature, light-demanding species with low wood density and small seeds) within edge habitats; and d) functional richness, evenness and divergence of dispersal traits are expected to decrease with increasing fragment isolation and decreasing matrix quality. Specifically, we expect abiotically-dispersed species and small-seeded, animal-dispersed species to dominate in more isolated fragments surrounded by a less hospitable matrix.

## Materials and methods

### Study area

The forest of the East Usambara Mountains stretches continuously from about 250 m to 1100 m asl in the southern part of this mountain range to form what is now protected as Amani Nature Reserve (8380 ha; -5˚04'58.80" S 38˚40'1.20" E). To the north of this reserve is Nilo Nature Reserve, and eastwards is the Derema corridor and several large fragments of lowland forest. Rainfall averages at 2000 mm per annum, falling largely from March to May and October to November; however, with the exception of January and February, precipitation is prevalent in most other months due to moisture carried across from the adjacent Indian Ocean [29]. The forest on the submontane plateau, in and around the primary study area of Amani Nature Reserve, is dominated by a suite of wet rainforest species. These include two emergent species *Newtonia buchananii* (Fabaceae) and *Maranthes goetzeniana* (Chrysobalanaceae), and several canopy and midstory/understorey species such as *Allanblackia stuhlmannii* (Clusiaceae), *Cephalosphaera usambarensis* (Myristicaceae), *Sorindeia madagascariensis* (Anacardiaceae), *Parinari excelsa* (Chrysobalanaceae), *Isoberlinia schefflerii* (Fabaceae), *Greenwayodendron suaveolens* (Annonaceae), *Anisophyllea obtusifolia* (Anisophylleaceae), *Leptonychia usambarensis* (Sterculiaceae), *Myrianthus holstii* (Urticaceae), *Macaranga capensis* (Euphorbiaceae), *Trilepisium madagascariense* (Moraceae) and *Strombosia scheffleri* (Olacaceae). The forest also contains *Maesopsis eminii* (Rhamnaceae), an exotic, invasive gap- and edge-specialist tree species [29,30].

Amani Nature Reserve is surrounded by several forest fragments of varying sizes in the submontane plateau (S1 Fig) and is primarily separated by a homogenous matrix of tea plantations. Apart from subsistence cultivation, which has shaped the forested landscape in more recent decades, much of the extensive forest loss and fragmentation arose from initial human occupation in the early pre-colonial period [31], but more extensively from the historical expansion of tea plantations, starting in the late 1800s [32]. Loss of original forest cover is estimated to exceed 50% [33]. Ten forest fragments and a large portion of the continuous forest were used to sample tree communities in 67 vegetation plots between May and July 2000; all sites are at 900–1100 m asl (S1 Table). Each vegetation plot was 20 x 20m and the plots were randomly located at ~25, ~150, ~250 and ~ 400 m from the forest edge towards the interior; smaller fragments (i.e. < 20 ha) did not have plots sampled at >200 m from the forest edge. All trees ≥ 10 cm Diameter at Breast Height (DBH) within each plot were identified to species.

## Functional trait data

We collated five traits (seed length, dispersal mode, shade tolerance, maximum tree height, and wood density) that correspond to key dimensions of species ecological strategies and have been previously used to explain competitive ability, growth, and reproduction in the context of forest fragmentation [14]. Traits for all the tree species were obtained through an exhaustive search of existing literature as well as online databases (S2 Table). Data on seed length and dispersal mode were obtained from Chapman et al. [34] and the African Tree Database (https://figshare.com/articles/Plant_animal_interactions_from_Africa/1526128). Seed length was based on the largest average length of the diaspore that is transported by the vector, and not necessarily the seed kernel size. Seed length reflects a seed number-seedling survival trade-off with small seeds being produced in large quantities and being better colonizers than larger seeds at the expense of withstanding lack of resources or different hazards thus reducing seedling survival and establishment [35]. Dispersal mode included zoochory (animal-dispersal), anemochory (wind-dispersal), and barachory (gravity or explosive dispersal). Dispersal mode influences the capacity of an individual to colonize newly formed or isolated fragments [36]. Maximum tree height was obtained from the literature [37] and an online database (http://www.prota.org). Maximum tree height is associated with competitive ability for light with taller trees displaying greater carbon assimilation potential than smaller trees [38–40]. Wood density for each species, or genus (when data for a species was not known), was derived from the global wood density database [41,42]. Wood density is a critical component for many essential functions, such as mechanical support and nutrient storage [43] and reflects a trade-off between radial growth to acquire physical stability at the expense of vertical growth [44,45]. Finally, for shade-tolerance guilds, we followed the classifications of Ouédraogo and collaborators [46,47], and where necessary, supplemented information from other sources [37,48]. Plant successional guilds included pioneer (species dependent on gaps or forest edge to establish), shade-tolerant (species dependent on shade across different ontogenetic levels), and light-demanding non-pioneer (species that establish in shade but initially require light to maximize growth) [46, sensu 49].

## Fragmentation metrics

Fragments were mapped using the high-resolution satellite imagery from Google Earth Pro. After mapping, metrics were calculated for each plot and all forest fragments using the GRASS GIS software [50,51]. Edge effects were evaluated based on the calculated distance of the center of the sampled plot to the forest edge. We also calculated fragment area ($km^2$), distance from the edge of a fragment to the closest edge of the continuous forest (m), matrix quality based on the surrounding cultivated land, and shape complexity.

For assessing matrix quality, we first characterized the matrix habitat of the study area into three land cover types: tea plantations (the primary matrix habitat), Eucalyptus woodlots, and subsistence cultivation of mixed crops such as bananas, beans, maize, cassava, cardamom, cloves and cinnamon. Tea and eucalyptus plantation represent a more hostile environment surrounding fragments, while subsistence cultivation is less hostile as it includes a mix of small forest patches and multi-crop species farms. To calculate matrix quality (MQ), we quantified the percentage of the forest edge in contact with each of the mentioned land covers as MQ = 100—(%tea + %eucalyptus). Thus, matrix quality is reduced as the percentage of forest edge in contact with the hostile matrix habitats comprising of tea or eucalyptus plantation increases.

To describe the shape complexity of the fragment, we calculated a fractal dimension index [52,53] where lower values correspond to regular shapes and higher ones to convoluted shapes,

as follows:

$$FDI = \frac{2 \ln \left(\frac{P}{4}\right)}{\ln A}$$

where *P* represents the perimeter and *A* the fragment area. FDI measures how much a shape deviates from a circumference thus excluding the effect of area on the edge complexity; if fragments are more complex, the perimeter increases and yields a higher fractal dimension.

## Analysis

**Functional diversity.** We used functional indices that capture three major components of functional diversity: functional richness (FRic) or the amount of niche space occupied by the community [54], evenness (FEve) or regularity of the distribution of species abundances in the functional space [55] and divergence (FDiv) or variance of species trait distribution in the trait space [54]. We calculated multivariate FRic, FEve and FDiv using all five functional traits, as studies have demonstrated that considering a single trait can lead to an oversimplification of results [e.g. 56]. All indices were calculated within the FD package [57] in R (Version 3.6.1; R Core Team 2019) and Gower distances were used to calculate functional distance between species pairs as it allows for the inclusion of both continuous and categorical traits. We calculated a weighted FEve index using the abundance of species (defined using number of stems) with different trait values within a community and an unweighted FEve index independent of species abundances. We then compared both weighted and unweighted FEve indices to improve the interpretation of this index as suggested by Legras and Gaertner [56]. To determine the functional trait value of a species community and explore how shifts along individual trait axes underlie the observed FRic, FEve and FDiv patterns, we calculated community-weighted mean (CWM) values (i.e. mean plot-level species trait values weighted by their relative abundance). To explore how community shifts along single traits underlie the observed FRic, FEve and FDiv patterns, we also calculated community-weighted mean (CWM) trait values (i.e. species trait values weighted by their relative abundance in each plot).

To further examine the potential drivers of functional diversity from the forest edge to interior of forests, we examined patterns of recruitment between edge and interior plots. It is possible that edge habitats have a more even distribution in wood density due to a mix of pre-fragmentation trees with high wood density and new post-fragmentation trees characterized by mostly low wood density. Thus, to evaluate whether there was a difference in tree size, we compared size distributions between interior and edge habitats for each guild using a Kolgomorov-Smirnov test. Edge plots were defined as all plots within <100 m of the fragment/forest edge and interior plots included all plots >100 m from the edge following Laurance [58] as 100 m being the threshold for edge effects.

**Statistical modelling.** We implemented a structural equation modelling approach using the R package piecewiseSEM [59] to investigate direct and indirect relationships of fragment- and landscape-scale variables in predicting local functional diversity. The general model investigated in this study (Fig 1) hypothesizes that variation in functional diversity among plots can be explained by the interacting direct and indirect effects of processes occurring at the fragment-level and the landscape-level. These include distance of plot to the nearest forest edge, distance of fragment to continuous forest as a measure of isolation, fragment area, matrix quality, and shape complexity. We used linear functions for all relationships in the structural equation models and ran separate models for each functional metric. To derive comparable estimates, we standardized all quantitative predictors to a mean of zero and standard deviation of one. In some cases, variables were log-transformed to achieve a normal error distribution.

To further explore potential shifts in trait distributions underlying variation in FRic FEve and FDiv, we used general linear models as a post-hoc test to determine changes in CWM traits values as a result of fragment- and landscape-scale variables. In these models, the responses are CWM values and the predictors are fragment- and landscape-level variables.

## Results

### Relationships among fragmentation variables

We found a strong positive relation between matrix quality and fragment isolation (coef = 0.704, se = 0.11, p < 0.001; Fig 2) and found a weak relationship between matrix quality and distance from plot to forest edge (coef = -0.595, se = 0.189, p = 0.568; Fig 2). The distance from plot to forest edge increased with fragment area (coef = 0.603, se = 0.156, p < 0.001; Fig 2) and tended to decrease as shape complexity of the fragment was reduced (coef = -0.423, se = 0.167, p = 0.015; Fig 2). Finally, shape complexity was positively associated with fragment area (coef = 0.541, se = 0.116, p < 0.001; Fig 2), with larger fragments characterized by more complex shapes than small fragments.

### Response of functional diversity to fragmentation

We found that direct and indirect effects between fragment-level fragmentation attributes appeared to be important drivers of local functional diversity within plots, while landscape-level attributes seemed to be less important in most cases. Functional richness tended to decrease with reduced shape complexity (coef = -0.936, se = 0.314, p = 0.004; Fig 2A). In addition, we found evidence that functional evenness tended to increase with fragment isolation (coef = 0.038, se = 0.017, p = 0.03; Fig 2B). No other significant effects at the landscape or fragment level were captured on functional evenness (Fig 2B). Finally, functional divergence tended to decrease with distance of a plot from the forest edge (coef = -0.028, se = 0.012, p = 0.031; Fig 2C), with no other significant effects on functional divergence (Fig 2C).

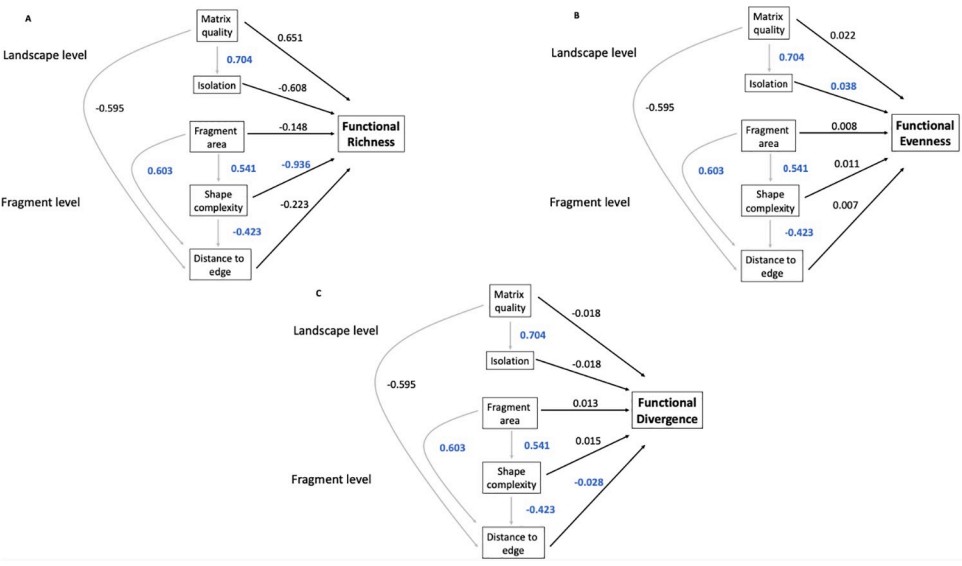

**Fig 2. Structural equation models examining the effects of forest fragmentation on functional diversity in the East Usambara Mountains, Tanzania.** A) functional richness, B) functional evenness and C) functional divergence. Grey lines represent indirect effects and dark lines representing direct effects. Values associated to lines represent standardized path coefficients. Significant results (p ≤ 0.05) are represented in dark blue.

## Shifts in community weighted mean traits with fragmentation

Values of community-weighted means were significantly associated with fragment isolation and distance from plot to forest edge (i.e. edge effects). CWM values for dispersal mode was significantly associated with fragment shape complexity, with anemochory increasing in more complex fragments (estimate = 0.182, se = 0.073). CWM values for wood density significantly declined with increasing fragment isolation (estimate = -0.015, se = 0.006). Furthermore, CWM values for successional guilds were significantly associated with fragment isolation and edge effects (distance from plot to forest edge). Specifically, CWM values for shade tolerance significantly declined with increasing fragment isolation (estimate = -0.071, se = 0.025) and increasing distance from plot to forest edge (estimate -0.388, se = 0.122). Distance from plot to forest edge was negatively associated with CMV values of light-demanding non-pioneer species (estimate = -0.171, se = 0.083), but, in contrast, positively associated with pioneer species (estimate = 0.097, se = 0.022). We also found distance of plot to forest edge was positively associated with maximum height (estimate = 1.305, se = 0.630) and wood density (estimate = 0.022, se = 0.005). See Table 1 for all results.

## Patterns of recruitment between edge and interior plots

We found significant differences in size distributions between the edges and the interior habitats for light-demanding non-pioneer species (D = 0.25, p = 0.03), with many small sized individuals found at the edges of the forest (Fig 3). We found a similar pattern for pioneer species (Fig 3), although this difference was not statistically significant (D = 0.18, p = 0.23). For shade-tolerant species, larger individuals were found at the interior of the forest (Fig 3), but, similar to pioneer species, this difference was not statistically significant (D = 0.07, p = 0.65).

## Discussion

Delineating the different processes that occur during landscape fragmentation and evaluating how they affect functional diversity is challenging. This is because landscape fragmentation leads to a series of changes in forest dynamics that occur at multiple spatial scales. Using a SEM approach with data from the rainforest of East Usambara Mountains, Tanzania, we detect several direct and indirect effects of forest fragmentation for different facets of functional diversity. We use this example to illustrate the great potential for significant advancements towards a more in depth understanding of the ecological consequences of forest fragmentation. At the landscape level, we find an indirect effect of matrix quality on functional evenness via its effect on increased fragment isolation. At the fragment-level, we find an indirect effect of fragment area on functional richness and functional divergence via its effects on shape complexity and edge effects (i.e. distance from plot center to forest edge), respectively. In this study, loss of shape complexity leads to significant changes in functional richness for traits related to dispersal mode. For resource use traits, we find that functional richness and divergence decline with decreasing shape complexity and distance from plot to forest edge, respectively, while functional evenness increased with isolation. Our results also suggested a negative relationship between fragment shape complexity and distance from plot center to forest edge, in line with previous work [7,60,61]. A reduction in fragment shape complexity might exacerbate edge effects on functional diversity, effects that might not be revealed when analysed using simple regression [7].

The relative importance of landscape and fragment-level factors vary considerably between traits, but fragment-level factors were generally more important than landscape characteristics in explaining variation in functional richness and divergence. By favouring tree species with fast growth, edge effects and shape complexity seem to be key drivers of changes in the

**Table 1. Effects of isolation, shape complexity and distance to edge on community weighted means for plant functional traits in the East Usambara Mountains, Tanzania.**

| Trait | Metric | Estimate | SE | t | p-value |
|---|---|---|---|---|---|
| **Anemochory** | Isolation | -0.064 | 0.078 | -0.812 | 0.424 |
| **Zoochory** | | -0.005 | 0.013 | -0.410 | 0.683 |
| **Barochory** | | 0.073 | 0.133 | 0.551 | 0.586 |
| **Height** | | -0.406 | 0.653 | -0.621 | 0.537 |
| **Light-demanding Non-Pioneer** | | 0.111 | 0.083 | 1.344 | 0.185 |
| **Pioneer** | | 0.179 | 0.121 | 1.482 | 0.145 |
| **Shade-Tolerant** | | -0.071 | 0.025 | -2.903 | 0.005 |
| **Seed length** | | -0.458 | 0.671 | -0.068 | 0.946 |
| **Wood density** | | -0.015 | 0.006 | 0.006 | 0.012 |
| **Anemochory** | Shape complexity | 0.182 | 0.073 | 2.476 | 0.020 |
| **Zoochory** | | 0.007 | 0.013 | 0.528 | 0.600 |
| **Barochory** | | 0.059 | 0.113 | 0.526 | 0.603 |
| **Height** | | -0.009 | 0.655 | -0.014 | 0.989 |
| **Light-demanding Non-Pioneer** | | 0.119 | 0.081 | 1.473 | 0.147 |
| **Pioneer** | | 0.038 | 0.122 | 0.309 | 0.759 |
| **Shade-Tolerant** | | 0.013 | 0.026 | 0.505 | 0.616 |
| **Seed length** | | -0.478 | 0.668 | -0.716 | 0.477 |
| **Wood density** | | 0.001 | 0.006 | 0.210 | 0.834 |
| **Anemochory** | Distance to edge | -0.006 | 0.104 | -0.059 | 0.953 |
| **Zoochory** | | -0.015 | 0.013 | -1.161 | 0.251 |
| **Barochory** | | 0.104 | 0.106 | 0.980 | 0.334 |
| **Height** | | 1.305 | 0.630 | 2.071 | 0.044 |
| **Light-demanding Non-Pioneer** | | -0.171 | 0.083 | -2.070 | 0.044 |
| **Shade-Tolerant** | | -0.388 | 0.122 | -3.176 | 0.003 |
| **Pioneer** | | 0.097 | 0.022 | 4.326 | < 0.001 |
| **Seed length** | | 1.127 | 0.652 | 1.727 | 0.09 |
| **Wood density** | | 0.022 | 0.005 | 4.249 | < 0.001 |

competitive hierarchies of tree communities in the fragmented forests of the East Usambaras. The lesser importance of landscape-scale variables in the current study, only observed for functional evenness, may be driven in part by our focus on plot-scale functional diversity. While the plot-scale data are informative in assessing variation in diversity at small spatial scales, they might not capture changes in diversity at large scales (e.g., at landscape scales or patterns in beta-diversity). Future work examining patterns in functional diversity aggregated at larger scales and across landscapes will extend the work presented here and better inform models of the main factors driving tree communities in fragmented forests.

## Functional diversity of resource use traits in response to fragmentation

A decrease in area of suitable habitat is expected to lead to the loss of species and a corresponding narrowing of trait value, resulting in lower alpha functional richness [62]. We found evidence of a decline in functional richness and divergence for traits related to resource use (e.g. wood density and regeneration strategy) due to reduced shape complexity and distance to edge respectively, both mediated by fragment area. Our results suggest the presence of strong post-fragmentation edge effects, leading to an increase in pioneer species while shade tolerant species are negatively impacted. In addition to area-based effects, fragmentation creates more

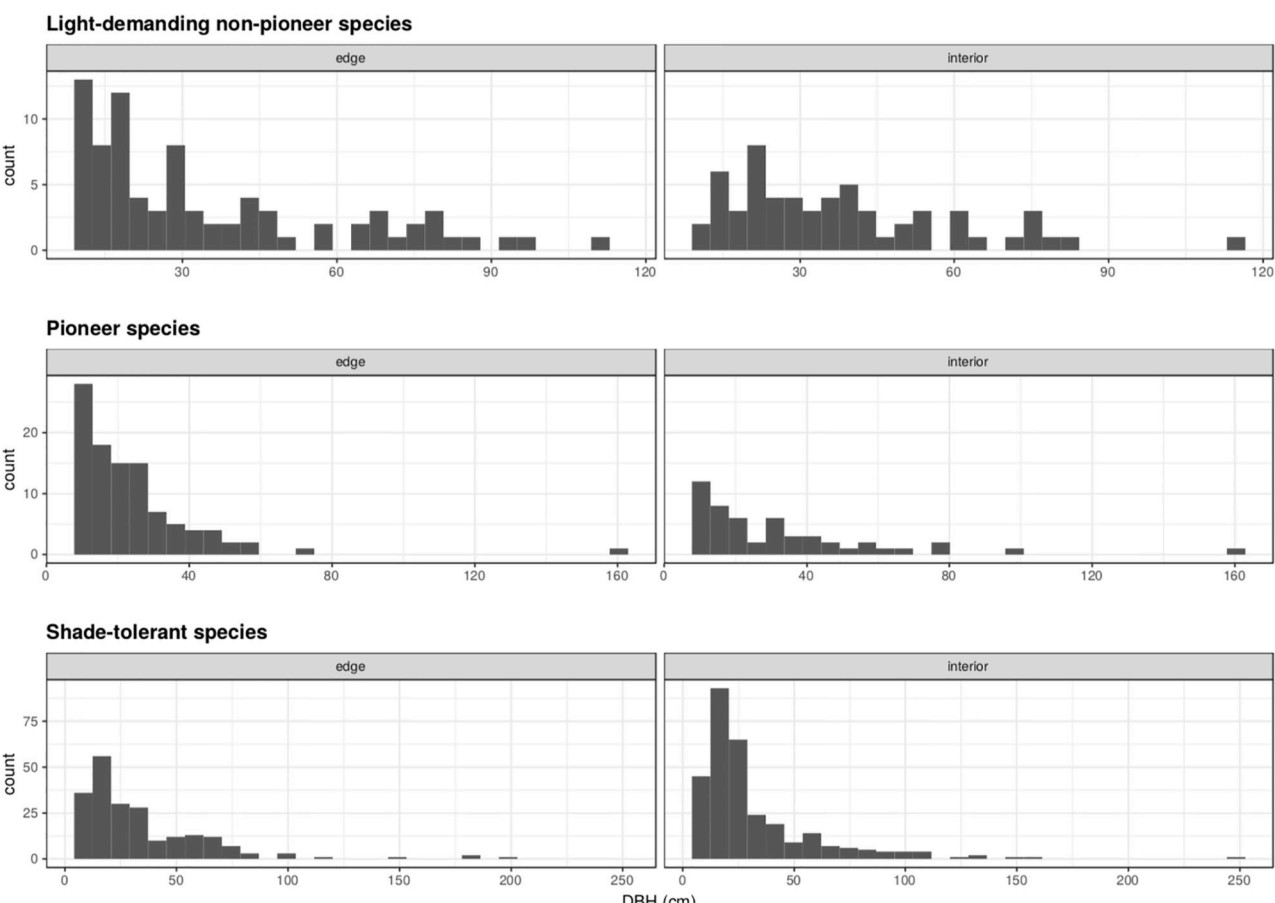

**Fig 3. Size distribution of each of the three successional guilds at edge versus interior vegetation plots in the East Usambara Mountains, Tanzania.**

edge habitat typified by elevated radiation, temperature and wind turbulence, and lower soil fertility and air moisture [63,64]. However, CWM trait values suggested that even if we found evidence of functional divergence decreasing with distance from plot to forest edge, species at the edges were more functionally diverse. This is likely the result of strong post-fragmentation edge effects leading to an increase of small stature, light-demanding species characterized by low wood density, combined with the older and taller, shade-tolerant and high wood-density species persisting from pre-fragmentation communities.

The studied area has experienced a long history of land use with varying levels of anthropogenic disturbance, resulting in significant forest loss and fragmentation [33]. In this highly fragmented landscape, edge effects on microclimate variables (air temperature, vapor pressure deficits and light intensity) are stronger within 60 to 94 m from the edge, as compared to the forest interior [65], explaining the observed changes in the competitive hierarchies of tree communities in the fragmented forest of the East Usambara Mountains. Functional divergence for resource use traits decreased with increasing distance of the plot from the forest edge, especially in large fragments as smaller fragments (< 20 ha) generally included far fewer interior plots (>200 m from the forest edge). The edge effects found here are in line with other studies [17,18,66–68], where species associated with slow growth rates are outcompeted in forest edges by light-demanding or pioneer species with fast growth. Old-growth species are particularly vulnerable to the detrimental effects of wind turbulence, desiccation, and liana

dominance that characterise the edge of forest fragments [15], including those in the East Usambaras [65].

## Functional diversity of dispersal traits in response to fragmentation

Fragments with more elongated shapes have higher proportion of total edge than interior habitat [60]. Reduced fragment shape complexity often results in low habitat heterogeneity, thus, communities in fragments with narrow and elongated shapes may exhibit reduced species richness and abundance [7]. Our results suggest that when fragments reach a certain reduced size, loss of shape complexity leads to significant changes in functional richness for traits related to dispersal mode. Increased complexity of fragment shape may limit impact of wind action to toppling large trees, which may explain why anemochorous species, like the emergent *Newtonia buchananni* (Fabaceae), remain in large and more complex East Usambara fragments. Small and less complex shaped fragments tend to be more vulnerable to edge-related wind damage increasing rates of windthrow and forest structural damage due to the higher ratio of perimeter to edge compared to larger and more complex shaped fragments [60]. Unfortunately, generalizing on the overall effects of fragment shape complexity on functional diversity is limited because it remains understudied compared to other fragmentation processes. It is important to highlight that our evidence of fragmentation effects on functional diversity comes from data of mature trees that represent the historical legacies of pre-fragmentation communities. Hence, without data on seedlings and saplings, it is difficult to ascertain whether the abundance of wind dispersed species is associated with reduced animal seed dispersers and therefore dispersal limitation which are negatively impacted by forest fragmentation in this study area [24,74,75]. Evidence from research in this study area has shown that several important frugivores are absent from or occur in lower abundance in forest fragments as compared to the continuous forest, threatening their persistence, as well as trees dependent on many these vectors [69–71].

Furthermore, we failed to uncover a relationship between traits related to dispersal (i.e. seed length, dispersal mode) and fragment isolation or matrix quality. Instead, our results suggest a less even distribution for traits related to resource use (e.g. wood density) as fragments became more isolated. Wood density is a strong indicator of successional dynamics with light wood often associated to early successional species (e.g. pioneer, light demanding species) that exhibit high fecundity and long-distance dispersal allowing them to colonize recently disturbed sites [41,72]. Therefore, light-wood species may be able to reach more isolated fragments perhaps due to better colonization abilities than hard-wood species. Fragmentation leads to increasing degree of isolation between fragments, hereby increasing the minimum dispersal distance for species from the regional pool to colonize fragments. However, it is important to highlight a potential correlation between dispersal and resource acquisition traits. Specifically, seed mass tends to define the mode of dispersal and is also related to the successional habit that determines resource acquisition strategies [40,73]. Light-demanding, early successional species often produce numerous small seeds and thus are considered better colonizers than shade-tolerant, late successional species [35], with potential to increase in abundance over time and negatively impacting the future establishment of late-successional trees.

The effects of isolation and matrix quality may require exploring functional diversity at a larger scale, beyond the local scale investigated in this study. Fragmentation leads to a high degree of isolation between the remaining fragments increasing the minimum dispersal distance for species from the regional pool to colonize fragments [61,74–76]. As species become more dispersal limited with decreasing fragment connectivity, we expect that fragments would become more similar in species composition, decreasing alpha functional richness and

increasing evenness. Moreover, a more diverse matrix may promote habitat heterogeneity between fragments increasing the range of total available niches at the landscape scale [77–79], potentially increasing functional richness and evenness. However, with most studies conducted at a local scale, the effects of isolation and matrix type on functional diversity remains fairly unexplored.

## Conclusion

By analysing trait variation due to processes occurring at the landscape scale and integrating this information with well-known fragment-scale processes using a structural equation approach, we were able to provide a more in-depth understanding of the different components of fragmentation and their impact on functional diversity. Specifically, the effects of fragment variables on functional diversity of trees were largely mediated by the indirect effect of fragment area on the amount of edge habitat and shape complexity. Our results demonstrate the power of this approach in detecting the effects of processes occurring at different spatial scales that may have been missed if only the direct impacts of landscape fragmentation would have been considered. This approach could greatly facilitate future empirical work in forest fragmentation and help advocate for management and restorations strategies that aim to achieve long-term persistence of remaining forests. Given that many fragmented forest systems will experience environmental conditions outside the range to which they are adapted, it is important to improve efforts to predict biodiversity responses to current human pressure to implement effective management and conservation strategies.

## Supporting information

**S1 Fig. Map of study area in the East Usambara Mountains of Tanzania.** Map of the study area in the East Usambara Mountains of Tanzania. The protected area, Amani Nature Reserve, includes the continuous forest (dark green) and the largest forest fragment (largest fragment in light green). The landcover classification (i.e. tea plantation, Eucalyptus plantation, forest and subsistence farming) was based on a Landsat-8 images from 2016 (courtesy of the U.S. Geological Survey) and performed using the random forest classification extension (r.learn. lm) in GRASS GIS.
(TIFF)

**S1 Table. Main fragment and continuous forest characteristics of study sites in the East Usambara Mountains, Tanzania.**
(PDF)

**S2 Table. Functional traits of tree species sampled in vegetation plots in the East Usambara Mountains, Tanzania.**
(PDF)

## Acknowledgments

Norbert Cordeiro and Henry Ndangalasi acknowledge the following for permits and assistance: Tanzania Commission for Science and Technology, East Usambara Conservation Area Management Programme, Amani Nature Reserve, East Usambara Tea Company, Tanga Regional Forest Office, Amani Parish and numerous individuals cited in Cordeiro *et al*. (2009). We are thankful to Sabine Kasel, Lionel Hertzog and an anonymous reviewer for valuable suggestions that greatly improved this manuscript.

## Author Contributions

**Conceptualization:** Jenny Zambrano, Noelle G. Beckman.

**Formal analysis:** Jenny Zambrano, Carol Garzon-Lopez, Lauren Yeager, Claire Fortunel, Noelle G. Beckman.

**Investigation:** Norbert J. Cordeiro, Henry J. Ndangalasi.

**Methodology:** Jenny Zambrano, Carol Garzon-Lopez, Lauren Yeager.

**Writing – original draft:** Jenny Zambrano.

**Writing – review & editing:** Norbert J. Cordeiro, Carol Garzon-Lopez, Lauren Yeager, Claire Fortunel, Noelle G. Beckman.

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
