## [Decision Letter · Decision Letter 0]

20 Feb 2020

PONE-D-19-34303

DISENTANGLING THE DIRECT AND INDIRECT EFFECTS OF FOREST FRAGMENTATION ON PLANT FUNCTIONAL DIVERSITY

PLOS ONE

Dear Dr Zambrano,

Thank you for submitting your manuscript to PLOS ONE. After careful consideration, we feel that it has merit but does not fully meet PLOS ONE’s publication criteria as it currently stands. Therefore, we invite you to submit a revised version of the manuscript that addresses the points raised during the review process.

You will find further below the comments by three reviewers. They have in common that further work is required on the manuscript, though their recommendations are different from each other. Each of the issues they bring up has merits, so I think they all need attention. Forest fragmentation is a complex process, and providing clues about factors and their interactions needs careful attention. I find the hints by the reviewers all relevant.

We would appreciate receiving your revised manuscript by Apr 05 2020 11:59PM. To enhance the reproducibility of your results, we recommend that if applicable you deposit your laboratory protocols in protocols.io, where a protocol can be assigned its own identifier (DOI) such that it can be cited independently in the future. For instructions see: http://journals.plos.org/plosone/s/submission-guidelines#loc-laboratory-protocols

We look forward to receiving your revised manuscript.

Kind regards,

Berthold Heinze

Academic Editor

PLOS ONE

Additional Editor Comments (if provided):

After some struggle (19 potential reviewers were contacted), we finally obtained three high quality reviews that help a lot in shaping this manuscript further into a very interesting contribution. The three reviews are divided in their recommendation, maybe because they go to different levels of depth, but for me the long and short of it is that some aspects require additional thoughts (or analyses), some of the wording needs careful reconsideration, and some technical aspects need attention (e.g. in some figures). I would like to invite the authors to take care of all issues mentioned by the reviewers and provide comment/action for all of these.

Journal Requirements:

2. In your Methods section, please provide additional location information of the study area, including geographic coordinates for the data set if available.

4. We note that Figure S1 in your submission contain map images which may be copyrighted. All PLOS content is published under the Creative Commons Attribution License (CC BY 4.0), which means that the manuscript, images, and Supporting Information files will be freely available online, and any third party is permitted to access, download, copy, distribute, and use these materials in any way, even commercially, with proper attribution. For these reasons, we cannot publish previously copyrighted maps or satellite images created using proprietary data, such as Google software (Google Maps, Street View, and Earth). For more information, see our copyright guidelines: http://journals.plos.org/plosone/s/licenses-and-copyright.

a.    You may seek permission from the original copyright holder of Figure S1 to publish the content specifically under the CC BY 4.0 license. 

5.Please include captions for your Supporting Information files at the end of your manuscript, and update any in-text citations to match accordingly. Please see our Supporting Information guidelines for more information: http://journals.plos.org/plosone/s/supporting-information.

Reviewers' comments:

Reviewer's Responses to Questions

**Comments to the Author**

1. Is the manuscript technically sound, and do the data support the conclusions?

Reviewer #1: Yes

Reviewer #2: Yes

Reviewer #3: Yes

2. Has the statistical analysis been performed appropriately and rigorously? 

Reviewer #1: Yes

Reviewer #2: Yes

Reviewer #3: Yes

3. Have the authors made all data underlying the findings in their manuscript fully available?

Reviewer #1: Yes

Reviewer #2: No

Reviewer #3: No

4. Is the manuscript presented in an intelligible fashion and written in standard English?

Reviewer #1: Yes

Reviewer #2: Yes

Reviewer #3: Yes

5. Review Comments to the Author

Reviewer #1: Zambrano et al. present an interesting work using a multi-scale SEM approach to tease apart the relative importance of suits of fragmentation effects on functional diversity in forest ecosystems. By exploring five key functional traits (i.e., seed length, dispersal mode, shade tolerance, maximum tree height and wood density), they found forest fragmentation directly and indirectly influences tree functional richness and evenness. Fragment area combing with edge habitat and shape complexity reduce functional richness and evenness for those traits related to resource acquisition as well as selecting fast grow tree species. Overall, I enjoy reading the work. The manuscript is well organized, and the framework, results and discussion are clear. This work is also timely important for forest management. I have only one major concern about quantifying functional diversity.

To account for unequal sampling effort, functional diversity indices were calculated through rarefied communities. It’s unclear for me about the rationale to use the mean 1000 random draws to represent the functional diversity indices. I am wondering whether the results for the expected functional indices are the same as the observed one. In addition, I cannot follow the rationale in Lines 230-234, too.

Minor concerns:

Line 112: What does UNESCO represent?

Line 248: Do you mean AICcs are less than 2?

Lines 299-307: This section can be moved to Methods section.

Line 476: Full journal name?

Reviewer #2: PONE_D_19_34303

GENERAL COMMENTS

A sound study that is well written with short-comings duly acknowledged by authors. My main comment focusses on the need for a greater consideration of animal vectors given that the majority of trees were animal dispersed (e.g. rather than wind dispersed). There was no mention on the habitat requirements/preferences on the animal vectors that would no doubt have an impact on plant colonisation (e.g. edge specialists vs core habitat requirements for fauna). More comments on this point provided below.

ABSTRACT

L25, ‘fragment … scale’, I assume you mean ‘local … scale’? (e.g. as referred to in L109)

INTRODUCTION

The introduction lacks any material on the role of animals in seed dispersal and plant colonisation (of both edges and forest interior). Given that there are very few wind-dispersed species in the forest and that nearly all are animal dispersed (including most of the pioneer species, from Table S1), there needs to be some consideration of fragmentation effects on colonisation by animal dispersed species – and I would also think some associated hypotheses?

L52, L58, missing space

L67, clarify what you mean by a ‘harsh matrix’

L78, still not clear what you mean by ‘fragment-scale’ (given fragment size can vary by orders of magnitude) – clarify so that this is clear in its subsequent use throughout

L93, passive sentence structure – revise

L114-125, there are many predictions included here – would it be possible to tighten up this paragraph? E.g. L115, 116, 123 each relate to pioneer species and their preference for edges.

L116, what about potential for long-distance dispersal via animal vectors?

L127, ‘fragment-scale fragmentation’ – meaning? (also L128)

MATERIALS AND METHODS

L144-148, for an international audience, it would be useful to include associated plant families (for those not familiar with the species)

L147, closing bracket here but there is no opening bracket

L157, 55 plots does not match the total number of plots shown in FigS1, nor the number of plots listed in Table S1.

L158, 20 × 20 m (not 20x20m)

L159, some fragments seem to have many more plots (Fig S1)?

L161, define DBH, measured for what? Presumable DBH and height only? How was height measured? Some indication of level of accuracy is needed.

L199-200, does this mean the metric considered the proportion of edge in contact with each of the different types of cultivated land? (e.g. given that each would potentially represent different habitat value for dispersal vectors, pollinators etc., differential effects on microclimate …)

RESULTS

Figure 3, from the scale shown on the map (Fig S1), I don’t know how distance to the continuous forest could be in the order of 1000’s of km (the scale shown in Fig S1 may also be incorrect).

DISCUSSION

L339-345, yes!

L367, avoid starting paragraph with ‘Furthermore’

L373, ‘compared’ not ‘compare’

L398, I don’t quite follow – wouldn’t it be easier to pick up a trend is you have a large range of seed dispersal modes? Moreover, the majority of the species were zoochorous – so perhaps it is the lack of diversity in dispersal modes that prevented detection of any trends? Zoochorous species may be ant dispersed, dispersed via ingestion or dispersed via adhesion – perhaps considering the finer levels of these dispersal modes would help given the flow on implications to dispersal distances and also habitat requirements of the animal vectors. [I see you make reference to this point in L405-407; however outside of gape-width, could you make finer groupings based on the type of animal vector – or are they all bird dispersed via ingestion?]

CONCLUSION

L421-428, this is introductory material.

SUPPLEMENTARY

Multiple font types used, inconsistent number of decimal places used -this needs tightening up.

Figure S1, Latitude and Longitude should be included along the edges of the map border. It looks like the scale legends are incorrect (possibly in both maps). What does the ‘white’ area represent? If the area between the forest fragments supports other land uses (e.g. tea plantations as indicated in text), then this should be shown.

Table S1, the number of plots listed in the table does not seem to match the plots shown in Figure S1? I wouldn’t describe this table as one of forest metrics as the only metric shown in Fragment Area. Do you need to differentiate the sizes according to small, medium, large (categorical attributes) given you have the actual area? Is the large green fragment considered one fragment? The numbers of fragments listed in the table don’t seem to match Figure S1.

Table S2, units are needed for Seed length, Height, Wood density.

Reviewer #3: In this manuscript Zambrano et al analyze the response of different functional diversity metrics to fragmentation variables separated into landscape and fragment-level using SEM. They report various responses of functional richness, evenness and divergence to fragmentation. The paper is interesting and generally well-written, my main suggestions for improvments are:

- check the consistency between the reported results and the statements in the discussion, for instance in the result a decrease in functional richness with distance to the edge is reported while in the discussion the opposite is stated

- the result section could be improved by separating the discussion of the inter-relation between the fragmentation variables from the discussion of the effect of fragmentation on functional diversity

Additionaly I have the following more detailed comments:

Line 37: this manuscript is about functional diversity, I recomend to change „community composition“ by „functional diversity“ or similar

Line 52: Replace „their local diversity“ by „local functional diversity“

Line 58: Unclear here what multivariate / univariate refer to, maybe drop it?

Line 74-89: Nice paragraph.

Line 95: One verb to much, choose one between identify and understanding.

Line 96-97: The interesting properties of SEM in this context could be made a bit clearer here. Something like: „may miss critical indirect effects between fragment-level and landscape-level fragmentation variables“

Line 107: I would be more specific here, the introduction focused on fragmentation only, „drivers“ is a bit vague here in that regard. Would suggest to replace with „fragmentation effects“.

Line 114-131: The hypothesis would need some greater consistency in their generality. For instance the first part „forest fragmentation is decreasing functional diversity“, is very vague given the objectives of the manuscript to disentagle direct and indirect relation across multiple scales. In this hypothesis, which aspect of forest fragmentation are you talking about? Which aspect of functional diversity? Similarly, the text under the (ii) subheaders is also very vague. I would recommend to only keep the three detailed points (matrix quality / isolation, fragment size and edge effects) separated into (i) to (iii).

Line 136: What range?

Line 138: Precipitation is usually given in mm

Line 214: remove „that describe“ and add a colon :

Line 222: How was abundance derived? Number of stems? DBH? Crown cover?

Line 226: This require clarification because above it is stated that all trees with >1m DBH were identified to species.

Line 244: Stepwise selection via AICc is an exploratory procedure that does not seem to fit to the confirmatory framework of the manuscript (clear hypothesis with predictions are set in the introduction). Why not just use the full model?

Line 260: I find the results as they stand now rather confusing because (at least) two different aspects are discussed at the same time: (i) the relation between the fragmentation variables (potential indirect effects) and (ii) the response of functional diversity to the fragmentation variables: I would recommend to separate the results into different subsections, in a first one the relations between the fragmentation variables would be discussed, and in subsequent ones the response of the different functional diversity metrics. Dropping the stepwise approach would also make the message clearer because now the inter-relation between the fragmentation variables is changing between the different functional diversity metrics which is a bit counter-intuitive. Working with a well thought-off full model would prevent this dispersion.

Line 285: „functional divergence tended to decrease“, with a p-value of 0.6, I find this a bit cheeky to give a direction to this effect that clearly could go in any (or no) direction.

Line 289: Please also report here the slopes and their p-value to be consistant with previous sections

Line 327: This is in contradiction with results reported line 266 for functional richness and in the result section I can see no reference to edge effects on functional eveness. Please check once again your result and derive the relevant conclusions from it!

Line 343-345: Good point! Maybe it is an option to already start looking at beta and gamma functional diversity in this manuscript?

Line 351-352: Again not supported by your results …

Table 1: Some of the coefficient estimates are very tiny (1e-20), why is that? What is the unit of your response variables? Did you consider re-scaling your response to improve model estimation (underflowing issue might arise with such tiny values)?

Figure 1: This figure is a bit confusing because in the SEM literature using a box (functional diversity) with arrows to different variables (functional richness …) is traditionally used to represent a latent variable. This is not the case here so to prevent confusion I would recommend to have one box with functional diversity and below it in bracket functional richness … Please also add in the legend that the different functional diversity metric were fitted in separate models.

Figure 2: Where is functional divergence? Also difference between significant / non-significant paths is not clear at all. Consider using a color scale. Also I am not sure that using line types (dashed, not dashed) to differentiate between direct and indirect effects is helpful here.

Figure S1: Some plots seem to be very close together, did you consider trying a Mantel test on the model residuals to check if there are any spatial autocorrelation present?

6. PLOS authors have the option to publish the peer review history of their article (what does this mean?). If published, this will include your full peer review and any attached files.

Reviewer #1: No

Reviewer #2: No

Reviewer #3: Yes: Lionel Hertzog

---

## [Author Response · Author response to Decision Letter 0]

5 Apr 2020

April 5, 2020

Dear Dr. Heinze,

We thank you for handling our original submission of our manuscript entitled “Investigating the direct and indirect effects of forest fragmentation on plant functional diversity” for consideration as a Research Paper in PlosOne as part of the Biodiversity Conservation Collection. Enclosed you will find a revised version of the manuscript that has been approved by my coauthors. We appreciate your consideration on the revised version.

In the following, we respond to each individual comment one-by-one. We hope you find the revisions satisfactory.

Sincerely,

Jenny Zambrano (on behalf of my co-authors)

Additional Editor Comments (if provided):

After some struggle (19 potential reviewers were contacted), we finally obtained three high quality reviews that help a lot in shaping this manuscript further into a very interesting contribution. The three reviews are divided in their recommendation, maybe because they go to different levels of depth, but for me the long and short of it is that some aspects require additional thoughts (or analyses), some of the wording needs careful reconsideration, and some technical aspects need attention (e.g. in some figures). I would like to invite the authors to take care of all issues mentioned by the reviewers and provide comment/action for all of these.

Journal Requirements:

Authors: Done

2. In your Methods section, please provide additional location information of the study area, including geographic coordinates for the data set if available.

Authors: We have included the main latitude and longitude coordinate in the main text and all GIS coordinates in the online database available at https://knb.ecoinformatics.org/view/doi:10.5063/F1KS6PX9

Authors: The data is available at: https://knb.ecoinformatics.org/view/doi:10.5063/F1KS6PX9

4. We note that Figure S1 in your submission contain map images which may be copyrighted. All PLOS content is published under the Creative Commons Attribution License (CC BY 4.0), which means that the manuscript, images, and Supporting Information files will be freely available online, and any third party is permitted to access, download, copy, distribute, and use these materials in any way, even commercially, with proper attribution. For these reasons, we cannot publish previously copyrighted maps or satellite images created using proprietary data, such as Google software (Google Maps, Street View, and Earth). For more information, see our copyright guidelines: http://journals.plos.org/plosone/s/licenses-and-copyright.

Authors: We have now use Landsat-8 images from 2016, courtesy of the U.S. Geological Survey. 

a. You may seek permission from the original copyright holder of Figure S1 to publish the content specifically under the CC BY 4.0 license. 

5.Please include captions for your Supporting Information files at the end of your manuscript, and update any in-text citations to match accordingly. Please see our Supporting Information guidelines for more information:http://journals.plos.org/plosone/s/supporting-information

¬ Authors: Done ________________________________________________________________________________________________________

Reviewers' comments:

Reviewer's Responses to Questions

Comments to the Author

Reviewer #1: 

Zambrano et al. present an interesting work using a multi-scale SEM approach to tease apart the relative importance of suits of fragmentation effects on functional diversity in forest ecosystems. By exploring five key functional traits (i.e., seed length, dispersal mode, shade tolerance, maximum tree height and wood density), they found forest fragmentation directly and indirectly influences tree functional richness and evenness. Fragment area combing with edge habitat and shape complexity reduce functional richness and evenness for those traits related to resource acquisition as well as selecting fast grow tree species. Overall, I enjoy reading the work. The manuscript is well organized, and the framework, results and discussion are clear. This work is also timely important for forest management. I have only one major concern about quantifying functional diversity.

Authors: Thank you for your support.

To account for unequal sampling effort, functional diversity indices were calculated through rarefied communities. It’s unclear for me about the rationale to use the mean 1000 random draws to represent the functional diversity indices. I am wondering whether the results for the expected functional indices are the same as the observed one. In addition, I cannot follow the rationale in Lines 230-234, too.

Authors: We removed the analysis using 1000 random draws of the rarefied community because all plants were exhaustively sampled in each plot. We used Community Weighted Means as a complement to functional diversity indices to better understand the functional shift observed in the studied area. 

Minor concerns:

Line 112: What does UNESCO represent?

Authors: We have written out UNESCO so all readers are aware of what it stands for.

Line 248: Do you mean AICcs are less than 2?

Authors: We have deleted this information since we used a full model and no model selection was required.

Lines 299-307: This section can be moved to Methods section.

Authors: Thank you for the suggestion; we moved part of this section to metods and left the results of this analysis to clarify our observations about distance to edge and, that complements the previous section.

Line 476: Full journal name?

Authors: Done

Reviewer #2: 

GENERAL COMMENTS

A sound study that is well written with short-comings duly acknowledged by authors. My main comment focusses on the need for a greater consideration of animal vectors given that the majority of trees were animal dispersed (e.g. rather than wind dispersed). There was no mention on the habitat requirements/preferences on the animal vectors that would no doubt have an impact on plant colonisation (e.g. edge specialists vs core habitat requirements for fauna). More comments on this point provided below.

Authors: Thank you for your suggestions. The concerns raised about animal seed dispersal are valid given that we did not clarify fully that we are evaluating functional diversity of adult tree communities post-fragmentation. We censused all trees > 10 cm DBH. The mature trees sampled in our census are largely remnants from pre-fragmentation, and for the most part, pioneer trees sampled in our census would have colonized post-fragmentation. Had we analysed functional diversity of early life history stages (i.e., seed, seedling, and saplings/recruits), we would have been able to make inferences about seed dispersal. 

ABSTRACT

L25, ‘fragment … scale’, I assume you mean ‘local … scale’? (e.g. as referred to in L109)

Authors: We replaced “fragment” by “local”

INTRODUCTION

The introduction lacks any material on the role of animals in seed dispersal and plant colonisation (of both edges and forest interior). Given that there are very few wind-dispersed species in the forest and that nearly all are animal dispersed (including most of the pioneer species, from Table S1), there needs to be some consideration of fragmentation effects on colonisation by animal dispersed species – and I would also think some associated hypotheses?

Authors: Thank you for your suggestions. However, as we previously mentioned we have analyzed mostly pre-fragmentation tree communities. Pioneers would be the main guild that have colonized post-fragmentation, and we have made this clearer throughout the Mss.

L52, L58, missing space

Authors: Corrected

L67, clarify what you mean by a ‘harsh matrix’

Authors: We mean a low quality matrix. We have included this clarification line 63.

L78, still not clear what you mean by ‘fragment-scale’ (given fragment size can vary by orders of magnitude) – clarify so that this is clear in its subsequent use throughout

Authors: We have clarified this in line 74.

L93, passive sentence structure – revise

Authors: Done

L114-125, there are many predictions included here – would it be possible to tighten up this paragraph? E.g. L115, 116, 123 each relate to pioneer species and their preference for edges.

Authors: We have completely rewritten the hypotheses to distinguish fragmentation effects on traits related to acquisition and traits associated with dispersal. Thank you for your suggestion.

L116, what about potential for long-distance dispersal via animal vectors?

Authors: As we previously mentioned we have analyzed mostly pre-fragmentation tree communities. In order, to determine the effects of long-distance dispersal we need to study seedling communities which was not the scope in this study.

L127, ‘fragment-scale fragmentation’ – meaning? (also L128)

Authors: We have removed “fragmentation” 

MATERIALS AND METHODS

L144-148, for an international audience, it would be useful to include associated plant families (for those not familiar with the species)

Authors: Thank you for the suggestion. We have included the associated plant families.

L147, closing bracket here but there is no opening bracket

Authors: We have removed the closing bracket 

L157, 55 plots does not match the total number of plots shown in FigS1, nor the number of plots listed in Table S1.

Authors: Corrected

L158, 20 × 20 m (not 20x20m)

Authors: Corrected

L159, some fragments seem to have many more plots (Fig S1)?

Authors: That’s correct. We already made this apparent in the text.

L161, define DBH, measured for what? Presumable DBH and height only? How was height measured? Some indication of level of accuracy is needed.

Authors: Corrected. In addition, we have removed “height” since it was not included in any of the analyses.

L199-200, does this mean the metric considered the proportion of edge in contact with each of the different types of cultivated land? (e.g. given that each would potentially represent different habitat value for dispersal vectors, pollinators etc., differential effects on microclimate …)

Authors: Here we mean matrix quality, and we have clarified in the methods (lines 209-217) how we define and calculate matrix quality based on the different types of cultivated land surrounding fragments. 

RESULTS

Figure 3, from the scale shown on the map (Fig S1), I don’t know how distance to the continuous forest could be in the order of 1000’s of km (the scale shown in Fig S1 may also be incorrect).

Authors: Corrected

DISCUSSION

L339-345, yes!

Authors: Thank you.

L367, avoid starting paragraph with ‘Furthermore’

Authors: Revised.

L373, ‘compared’ not ‘compare’

Authors: Corrected.

L398, I don’t quite follow – wouldn’t it be easier to pick up a trend is you have a large range of seed dispersal modes? Moreover, the majority of the species were zoochorous – so perhaps it is the lack of diversity in dispersal modes that prevented detection of any trends? Zoochorous species may be ant dispersed, dispersed via ingestion or dispersed via adhesion – perhaps considering the finer levels of these dispersal modes would help given the flow on implications to dispersal distances and also habitat requirements of the animal vectors. [I see you make reference to this point in L405-407; however outside of gape-width, could you make finer groupings based on the type of animal vector – or are they all bird dispersed via ingestion?]

Authors: Thank you. This is a good point. Unfortunately, we do not have data on the finer-scale categories of dispersal mode. We have altered this section as the new results include an effect of fragmentation on anemochory discussed in lines 401-406, and in addition, we clarify earlier that we are analyzing post-fragmentation communities of mature trees, much of which are remnants of pre-fragmentation, and hence changes in the diversity of dispersal modes are driven by pioneer species.

CONCLUSION

L421-428, this is introductory material.

Authors: We have deleted part of this section and move part of it to the end of the conclusion to reflect the important of using an approach such as the one we use in this study to better understand fragmentation effects and apply it to management efforts.

SUPPLEMENTARY

Multiple font types used, inconsistent number of decimal places used -this needs tightening up.

Authors: Corrected.

Figure S1, Latitude and Longitude should be included along the edges of the map border. It looks like the scale legends are incorrect (possibly in both maps). What does the ‘white’ area represent? If the area between the forest fragments supports other land uses (e.g. tea plantations as indicated in text), then this should be shown.

Authors: Corrected.

Table S1, the number of plots listed in the table does not seem to match the plots shown in Figure S1? I wouldn’t describe this table as one of forest metrics as the only metric shown in Fragment Area. 

Authors: Corrected.

Do you need to differentiate the sizes according to small, medium, large (categorical attributes) given you have the actual area? 

Author: We have deleted this column. 

Is the large green fragment considered one fragment? The numbers of fragments listed in the table don’t seem to match Figure S1.

Authors: Corrected.

Table S2, units are needed for Seed length, Height, Wood density.

Authors: Corrected.

Reviewer #3: In this manuscript Zambrano et al analyze the response of different functional diversity metrics to fragmentation variables separated into landscape and fragment-level using SEM. They report various responses of functional richness, evenness and divergence to fragmentation. 

Authors: Thank you for your support.

The paper is interesting and generally well-written, my main suggestions for improvments are:

- check the consistency between the reported results and the statements in the discussion, for instance in the result a decrease in functional richness with distance to the edge is reported while in the discussion the opposite is stated

Authors: Done.

- the result section could be improved by separating the discussion of the inter-relation between the fragmentation variables from the discussion of the effect of fragmentation on functional diversity

Authors: Thank you for the suggestion. We have divided the results from the SEMs into two sections: 1) the inter-relation between fragment variables and 2) effects on functional diversity.

Additionaly I have the following more detailed comments:

Line 37: this manuscript is about functional diversity, I recomend to change „community composition“ by „functional diversity“ or similar

Authors: Corrected.

Line 52: Replace „their local diversity“ by „local functional diversity“

Authors: Done

Line 58: Unclear here what multivariate / univariate refer to, maybe drop it?

Authors: Done.

Line 74-89: Nice paragraph.

Authors: Thank you!

Line 95: One verb to much, choose one between identify and understanding.

Authors: Corrected.

Line 96-97: The interesting properties of SEM in this context could be made a bit clearer here. Something like: „may miss critical indirect effects between fragment-level and landscape-level fragmentation variables“

Authors: Corrected.

Line 107: I would be more specific here, the introduction focused on fragmentation only, „drivers“ is a bit vague here in that regard. Would suggest to replace with „fragmentation effects“.

Authors: Corrected.

Line 114-131: The hypothesis would need some greater consistency in their generality. For instance the first part „forest fragmentation is decreasing functional diversity“, is very vague given the objectives of the manuscript to disentagle direct and indirect relation across multiple scales. In this hypothesis, which aspect of forest fragmentation are you talking about? Which aspect of functional diversity? Similarly, the text under the (ii) subheaders is also very vague. I would recommend to only keep the three detailed points (matrix quality / isolation, fragment size and edge effects) separated into (i) to (iii).

Authors: We have included the information of how functional diversity was defined and the expected results on diversity of traits related to resource acquisition and dispersal, as well as, the expected interactions between fragment- and landscape-level processes.

Line 136: What range?

Authors: Clarified.

Line 138: Precipitation is usually given in mm

Authors: Corrected.

Line 214: remove „that describe“ and add a colon :

Authors: Done

Line 222: How was abundance derived? Number of stems? DBH? Crown cover?

Authors: Number of stems. We have included this information.

Line 226: This require clarification because above it is stated that all trees with >1m DBH were identified to species.

Authors: Corrected

Line 244: Stepwise selection via AICc is an exploratory procedure that does not seem to fit to the confirmatory framework of the manuscript (clear hypothesis with predictions are set in the introduction). Why not just use the full model?

Authors: Thank you for the recommendation. We now present and discuss the results from full models.

Line 260: I find the results as they stand now rather confusing because (at least) two different aspects are discussed at the same time: (i) the relation between the fragmentation variables (potential indirect effects) and (ii) the response of functional diversity to the fragmentation variables: I would recommend to separate the results into different subsections, in a first one the relations between the fragmentation variables would be discussed, and in subsequent ones the response of the different functional diversity metrics. Dropping the stepwise approach would also make the message clearer because now the inter-relation between the fragmentation variables is changing between the different functional diversity metrics which is a bit counter-intuitive. Working with a well thought-off full model would prevent this dispersion.

Authors: Thank you for the recommendations. We have now divided the results into two sections and drop the stepwise approach.

Line 285: „functional divergence tended to decrease“, with a p-value of 0.6, I find this a bit cheeky to give a direction to this effect that clearly could go in any (or no) direction.

Authors: Following reviewer suggestions on analysis, we now report new results pertaining to functional divergence that show a strong relationship with shape complexity.

Line 289: Please also report here the slopes and their p-value to be consistant with previous sections

Authors: Done.

Line 327: This is in contradiction with results reported line 266 for functional richness and in the result section I can see no reference to edge effects on functional eveness. Please check once again your result and derive the relevant conclusions from it!

Authors: We have corrected this.

Line 343-345: Good point! Maybe it is an option to already start looking at beta and gamma functional diversity in this manuscript?

Authors: Thank you for your recommendation. This is something we tried in a previous analysis; however, due to the low replication at the landscape level and variation in sampling effort we were not able to draw strong conclusions from this analysis.

Line 351-352: Again not supported by your results …

Authors: Corrected.

Table 1: Some of the coefficient estimates are very tiny (1e-20), why is that? What is the unit of your response variables? Did you consider re-scaling your response to improve model estimation (underflowing issue might arise with such tiny values)?

Authors: Thank you for your suggestion. We have rescaled the CWM values.

Figure 1: This figure is a bit confusing because in the SEM literature using a box (functional diversity) with arrows to different variables (functional richness …) is traditionally used to represent a latent variable. This is not the case here so to prevent confusion I would recommend to have one box with functional diversity and below it in bracket functional richness … Please also add in the legend that the different functional diversity metric were fitted in separate models.

Authors: Corrected.

Figure 2: Where is functional divergence? Also difference between significant / non-significant paths is not clear at all. Consider using a color scale. Also I am not sure that using line types (dashed, not dashed) to differentiate between direct and indirect effects is helpful here.

Authors: Corrected and we have included a figure including functional divergence.

Figure S1: Some plots seem to be very close together, did you consider trying a Mantel test on the model residuals to check if there are any spatial autocorrelation present?

Authors: Thank you for the suggestion; however, we believe that the measurements of fragment and landscape properties (i.e. distance of plot to edge of fragment, distance of fragment to continuous forest), included in this study, take into count the spatial context of the focal units. We do include information on distance of plot to fragment edge and distance of fragment to forest, so we would expect this would take into account some of the spatial signature. Of course, we are not accounting for variation in variables such soil and topography, to give just two examples, hence we cannot be completely certain of the spatial signature stems from the spatial fragmentation processes or the underlying topography/soil.

---

## [Decision Letter · Decision Letter 1]

5 May 2020

PONE-D-19-34303R1

Investigating the direct and indirect effects of forest fragmentation on plant functional diversity

PLOS ONE

Dear Dr Zambrano,

Thank you for submitting your manuscript to PLOS ONE. After careful consideration, we feel that it has merit but does not fully meet PLOS ONE’s publication criteria as it currently stands. Therefore, we invite you to submit a revised version of the manuscript that addresses the points raised during the review process.

A few remaining suggestions of the reviewers are listed below. I think they will improve the manuscript further, so it is worth the extra round.

We would appreciate receiving your revised manuscript by Jun 19 2020 11:59PM. To enhance the reproducibility of your results, we recommend that if applicable you deposit your laboratory protocols in protocols.io, where a protocol can be assigned its own identifier (DOI) such that it can be cited independently in the future. For instructions see: http://journals.plos.org/plosone/s/submission-guidelines#loc-laboratory-protocols

We look forward to receiving your revised manuscript.

Kind regards,

Berthold Heinze

Academic Editor

PLOS ONE

Additional Editor Comments (if provided):

All reviewers are impressed by the additional work done for improving the manuscript. It is considered (almost) ready for publication. I think the few remaining points can easily be addressed (some may just be a matter of re-wording in order to gain clarity in expression); e.g. the hypotheses are mentioned by two of the reviewers. Also carefully consider the other comments please.

Reviewers' comments:

Reviewer's Responses to Questions

**Comments to the Author**

1. If the authors have adequately addressed your comments raised in a previous round of review and you feel that this manuscript is now acceptable for publication, you may indicate that here to bypass the “Comments to the Author” section, enter your conflict of interest statement in the “Confidential to Editor” section, and submit your "Accept" recommendation.

Reviewer #1: All comments have been addressed

Reviewer #2: (No Response)

Reviewer #3: (No Response)

2. Is the manuscript technically sound, and do the data support the conclusions?

Reviewer #1: Yes

Reviewer #2: Yes

Reviewer #3: Partly

3. Has the statistical analysis been performed appropriately and rigorously? 

Reviewer #1: Yes

Reviewer #2: Yes

Reviewer #3: Yes

4. Have the authors made all data underlying the findings in their manuscript fully available?

Reviewer #1: Yes

Reviewer #2: Yes

Reviewer #3: No

5. Is the manuscript presented in an intelligible fashion and written in standard English?

Reviewer #1: Yes

Reviewer #2: Yes

Reviewer #3: Yes

6. Review Comments to the Author

Reviewer #1: I've read the revised manuscript. The authors have done an excellent work. All of my concerns have been addressed. Congratulations!

Reviewer #2: GENERAL

A much improved manuscript. The authors have diligently addressed the numerous reviewer comments. I only have a few minor comments.

My comments relate to the unannotated version.

INTRODUCTION

L65, Are you suggesting that animal-dispersed species are highly dispersed due to their smaller seed? Please clarify – many animal-dispersed species have large seed and are dispersed long distances by being carried by animals (either internally / externally). L120 mentions abiotically-dispersed species with smaller seeds – to I suspect the sentence in L65 just needs to be rephrased for clarity.

L109-113, the hypothesis is poorly expressed – please clarify. I assume you are expecting a decline in functional richness, evenness and divergence?

MATERIALS AND METHODS

L255, space needed (100 m, not 100m)

RESULTS

L303-317, given the results are provided in Table 1, I see no need to list estimate and se in the text (the table could also include t and p).

L324-332, I could not find Figure 3 in the revised version?

DISCUSSION

L383, long ‘history’

ACKNOWLEDGEMENTS

Manuscript also improved by suggestions by reviewers.

SUPPORTING INFORMATION

Much better.

Reviewer #3: First of all, I would like to congratulate the authors for their impresive work in taking into account the large number of comments that were made in the previous version of the manuscript.

I have still two major issues with the manuscript as it now stands:

- Hypothesis: Hypothesis 1 is very long and rather hard to follow. I would recommend to split it up, for instance into a) functional richness, evenness and divergence of resource use traits are expected to decline with …., b) low matrix quality is expected to lead to stronger decline in …., c) trait distribution is expected to become more skewed towards …, d) functional richness, evenness and divergence of dispersal trais are expected to … I would switch hypothesis 2 and 1, hypothesis 2 present how you expect fragmentation effects operating at different scale to interact with each other. Hypothesis 2 is therefore more general. In hypothesis 2 you write: „distance to fragment edge tend to increase in small … fragments“, how is that possible? In smaller fragments every single points should be closer to the edge than in larger fragments.

- Discussion: Fragmentation and especially reduction in habitat area is a process that is usually expected to lead to declining biodiversity, yet looking at Figure 2 it seems that this is not the case here. Based on this main results I am missing a more explicit discussion of the absence of negative direct and indirect effect of reduced fragment area on functional diversity. This sounds rather provocative but your results tend to show that smaller forest patches do not have lower functional richness and evenness compared to larger patches but they even have higher functional divergence. I think that such discussion would be particularly interesting around the lines 368-369, where the opposite is expected. I think that this manuscript would be greatly enhanced by further interpretation of these results and their potential implications.

Minor comments:

- line 230: replace „determine“, maybe use „measure“ instead

- line 259: which package was used to compute the SEMs?

- line 309-312: does the decrease of CWM for shade tolerance with distance to forest edge means that plants are more shade-tolerant closer to the edge? This sounds rather counter-intuitive, do you have some explanation for this?

7. PLOS authors have the option to publish the peer review history of their article (what does this mean?). If published, this will include your full peer review and any attached files.

Reviewer #1: No

Reviewer #2: Yes: Sabine Kasel

Reviewer #3: Yes: Lionel Hertzog

---

## [Author Response · Author response to Decision Letter 1]

14 May 2020

Dear Dr. Heinze, 

We thank you for your handling of our previous submission to PlosONE entitled: Investigating the direct and indirect effects of forest fragmentation on plant functional diversity (ID: PONE-D-19-3430R1). The manuscript received two detailed expert reviews. There were a few remaining points that needed to be addressed leading the decision of minor revision.

Detailed responses can be found below. The suggestions made by the reviewers greatly helped to improve the work and the manuscript. 

We thank you again for your consideration and we hope you will find this new version of the manuscript suitable for publication in PlosONE.

Sincerely,

Jenny Zambrano

Additional Editor Comments (if provided):

All reviewers are impressed by the additional work done for improving the manuscript. It is considered (almost) ready for publication. I think the few remaining points can easily be addressed (some may just be a matter of re-wording in order to gain clarity in expression); e.g. the hypotheses are mentioned by two of the reviewers. Also carefully consider the other comments please.

Authors: Thank you for handling the paper.

Reviewer #1: I've read the revised manuscript. The authors have done an excellent work. All of my concerns have been addressed. Congratulations!

Authors: Thank you for your support and for your previous suggestions.

Reviewer #2: GENERAL

A much improved manuscript. The authors have diligently addressed the numerous reviewer comments. I only have a few minor comments.

My comments relate to the unannotated version.

Authors: Thank you for your support and for your previous suggestions.

INTRODUCTION

L65, Are you suggesting that animal-dispersed species are highly dispersed due to their smaller seed? Please clarify – many animal-dispersed species have large seed and are dispersed long distances by being carried by animals (either internally / externally). L120 mentions abiotically-dispersed species with smaller seeds – to I suspect the sentence in L65 just needs to be rephrased for clarity.

Authors: Corrected

L109-113, the hypothesis is poorly expressed – please clarify. I assume you are expecting a decline in functional richness, evenness and divergence?

Authors: Corrected

MATERIALS AND METHODS

L255, space needed (100 m, not 100m)

Authors: Corrected

RESULTS

L303-317, given the results are provided in Table 1, I see no need to list estimate and se in the text (the table could also include t and p).

Authors: Corrected

L324-332, I could not find Figure 3 in the revised version?

Authors: We had included Figure 3 in the revised version. 

DISCUSSION

L383, long ‘history’

Authors: Corrected

ACKNOWLEDGEMENTS

Manuscript also improved by suggestions by reviewers.

Authors: Done

SUPPORTING INFORMATION

Much better.

Authors: Thank you.

Reviewer #3: First of all, I would like to congratulate the authors for their impresive work in taking into account the large number of comments that were made in the previous version of the manuscript.

Authors: Thank you for your support and for your previous suggestions.

I have still two major issues with the manuscript as it now stands:

- Hypothesis: Hypothesis 1 is very long and rather hard to follow. I would recommend to split it up, for instance into a) functional richness, evenness and divergence of resource use traits are expected to decline with …., b) low matrix quality is expected to lead to stronger decline in …., c) trait distribution is expected to become more skewed towards …, d) functional richness, evenness and divergence of dispersal trais are expected to … I would switch hypothesis 2 and 1, hypothesis 2 present how you expect fragmentation effects operating at different scale to interact with each other. Hypothesis 2 is therefore more general. In hypothesis 2 you write: „distance to fragment edge tend to increase in small … fragments“, how is that possible? In smaller fragments every single points should be closer to the edge than in larger fragments.

Authors: Thank you for the suggestion. We have split the presentation of the hypothesis accordingly. We have also clarified and replaced “distance to fragment edge tend to decrease in small…fragments”.

- Discussion: Fragmentation and especially reduction in habitat area is a process that is usually expected to lead to declining biodiversity, yet looking at Figure 2 it seems that this is not the case here. Based on this main results I am missing a more explicit discussion of the absence of negative direct and indirect effect of reduced fragment area on functional diversity. This sounds rather provocative but your results tend to show that smaller forest patches do not have lower functional richness and evenness compared to larger patches but they even have higher functional divergence. I think that such discussion would be particularly interesting around the lines 368-369, where the opposite is expected. I think that this manuscript would be greatly enhanced by further interpretation of these results and their potential implications.

Authors: Larger fragments (area?) tended to include more interior plots (>200 m from the forest edge), whereas smaller fragments (i.e. < 20 ha) generally included far fewer interior plots due to their smaller size. The observed decrease in functional divergence may be explained by the fact that interior plots showed higher functional divergence than plots located closer to the edge. The decrease in functional divergence might have not been so significant in smaller fragments because most studied plots were found within 100m from the forest edges. We have clarified this in the Discussion (494-497).

Minor comments:

- line 230: replace „determine“, maybe use „measure“ instead

Authors: Corrected

- line 259: which package was used to compute the SEMs?

Authors: Corrected

- line 309-312: does the decrease of CWM for shade tolerance with distance to forest edge means that plants are more shade-tolerant closer to the edge? This sounds rather counter-intuitive, do you have some explanation for this?

Authors: Shade tolerance significantly declined with increasing distance from the plot to forest edge (see Table 1), while we found an increase in light-demanding species characteristic of edge habitats (i.e. pioneers and light-demanding non-pioneers).

---

## [Editor Report · Decision Letter 2]

11 Jun 2020

Investigating the direct and indirect effects of forest fragmentation on plant functional diversity

PONE-D-19-34303R2

Dear Dr. Zambrano,

We’re pleased to inform you that your manuscript has been judged scientifically suitable for publication and will be formally accepted for publication once it meets all outstanding technical requirements.

Kind regards,

Berthold Heinze

Section Editor

PLOS ONE

Additional Editor Comments (optional):

My apologies for the long time it took me to handle this manuscript. I have now gone through all the changes and answers, it all makes good sense to me and I am happy to accept this manuscript, which I am sure will make an outstanding contribution in this journal.
---

## [Editor Report · Acceptance letter]

19 Jun 2020

PONE-D-19-34303R2 

Investigating the direct and indirect effects of forest fragmentation on plant functional diversity 

Dear Dr. Zambrano:

I'm pleased to inform you that your manuscript has been deemed suitable for publication in PLOS ONE. Congratulations! Your manuscript is now with our production department. 

Kind regards, 

on behalf of

Dr. Berthold Heinze 

Section Editor

PLOS ONE